# Mohangic Acid H and Mohangiol: New *p*-Aminoacetophenone Derivatives from a Mudflat-Derived *Streptomyces* sp.

**DOI:** 10.3390/md23080307

**Published:** 2025-07-30

**Authors:** Juwan Son, Ju Heon Lee, Yong-Joon Cho, Kyuho Moon, Munhyung Bae

**Affiliations:** 1College of Pharmacy, Gachon University, Incheon 21936, Republic of Korea; ths3120@gachon.ac.kr (J.S.); jjuis@naver.com (J.H.L.); 2Department of Molecular Bioscience, Kangwon National University, Chuncheon 24341, Republic of Korea; yongjoon@kangwon.ac.kr; 3Multidimensional Genomics Research Center, Kangwon National University, Chuncheon 24341, Republic of Korea; 4College of Pharmacy, Kyung Hee University, Seoul 02447, Republic of Korea

**Keywords:** *Streptomyces* sp. AWH31-250, tidal mudflat, *p*-aminoacetophenone, computational DP4+ calculations, biosynthetic pathway, bioinformatic analysis, *Candida albicans* isocitrate lyase

## Abstract

*Streptomyces* sp. AWH31-250, isolated from a tidal mudflat in the Nakdong River estuary in Busan, Republic of Korea, was found to produce two novel *p*-aminoacetophenone derivatives, mohangic acid H (**1**) and mohangiol (**2**). Their planar structures were established by comprehensive 1D and 2D NMR spectroscopy, mass spectrometry, and UV analysis, possessing a shorter carbon-chain with a diene moiety, whereas known mohangic acids A–F bear a longer carbon-chain with a triene moiety. The absolute configurations of the key stereogenic centers were determined via computational DP4+ calculations and bioinformatic analysis of the ketoreductase domain sequence from the biosynthetic gene cluster. Based on the careful gene analysis along with whole-genome sequencing, the first plausible biosynthetic pathway of mohangic acids A–G and mohangiol was proposed. Mohangic acid H (**1**) and mohangiol (**2**) displayed moderate inhibitory activity against *Candida albicans* isocitrate lyase with IC_50_ values of 21.37 and 21.12 µg/mL, respectively.

## 1. Introduction

Marine intertidal mudflats constitute dynamic and unique ecosystems characterized by extreme environmental gradients in salinity, light exposure, temperature, oxygen level, and nutrient availability [1,2]. These variable conditions and the complex nature of mudflats promote exceptional biological and chemical diversity, leading to adaptive strategies, genomic evolution, and specialized metabolism in resident microorganisms, particularly actinomycetes [3]. Actinomycetes have been well-known as novel natural product producers harboring multiple polyketide synthase (PKS) and non-ribosomal peptide synthetase (NRPS) gene clusters [4]. The constant physiological challenges coupled with environmental stress in mudflats drive actinomycetes to extend their capacity for the biosynthesis of structurally complex secondary metabolites with diverse biological activities [5,6].

As part of our efforts to discover new bioactive natural products from marine mudflats, we isolated actinomycete strains from various intertidal mudflats in the Republic of Korea and investigated their bioactive secondary metabolites. Our previous studies successfully resulted in the discovery of new antibacterial flavonoid-type glycosides, the actinoflavosides [7], potent antibacterial cyclic depsipeptides, the hormaomycins [8], antifungal dilactone-tethered pseudo-dimeric peptides, the mohangamides [9], and an antiangiogenic pyrazolone-bearing peptide, WS9326H [10].

Among the diverse metabolites produced by mudflat-derived actinomycetes, mohangic acids represent polyketide derivatives characterized by a *p*-aminoacetophenone scaffold [11,12]. In our previous report, we briefly suggested a structural relationship between specific fragments of mohangic acids and candicidin D, a polyene macrolide antifungal agent, but the precise biosynthetic relevance remained unexplored [13]. Further chemical and genomic analysis of the actinomycete strain AWH31-250, isolated from a tidal mudflat in the Nakdong River estuary in Busan, Republic of Korea, led to the discovery of new derivatives of mohangic acids bearing a diene moiety, based on UV and MS data analysis. A large-scale fermentation of the strain and further purification yielded a new series of *p*-aminoacetophenone-derived metabolites, mohangic acid H (**1**) and mohangiol (**2**) (Figure 1).

Herein, we report the isolation, structural elucidation, biosynthetic pathway, and biological activities of two new mohangic acid derivatives, mohangic acid H (**1**) and mohangiol (**2**). 

## 2. Results and Discussion

### 2.1. Structural Elucidation

Mohangic acid H (**1**) was isolated as a yellow gum possessing the molecular formula C_24_H_33_NO_6_ based on ^1^H and ^13^C NMR data and high-resolution ESI mass spectroscopy. The ^1^H NMR spectrum of **1** in pyridine-*d*_5_ exhibited two overlapped aromatic protons at *δ*_H_ 8.26 and 8.08 ppm, four olefinic protons from 6.28 to 5.76 ppm, two carbinol protons at 4.62 and 3.50 ppm, seven aliphatic protons between 3.53 and 1.65 ppm, four methyl groups at 2.18, 1.19, 1.18, and 1.17 ppm in the shielded region of the spectrum. The ^13^C NMR and HSQC spectra of **1** revealed the presence of three carbonyl carbons at 199.7, 171.7 and 169.8 ppm, 10 sp^2^ carbons between 171.7 and 119.2 ppm, two oxygenated sp^3^ carbons at 79.1 and 72.8 ppm, five aliphatic sp^3^ carbons between 43.2 and 34.6 ppm, and four methyl carbons at *δ*_C_ 24.4, 18.4, 16.2 and 14.7 ppm. The molecular formula of **1** indicated nine degrees of unsaturation. Since ten olefinic carbon signals (corresponding to five double bonds) along with two carbonyl carbons account for seven degrees of unsaturation, the structure of mohangic acid H (**1**) should incorporate additional structure elements to account for the remaining two degrees. Further analysis of the UV absorption maximum at 270 nm indicated the presence of a conjugated system involving the *p*-aminoacetophenone moiety with a ketone group.

**Figure 1 marinedrugs-23-00307-f001:**
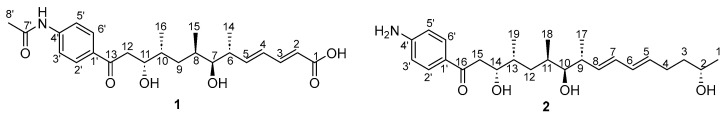
The structure of mohangic acid H (**1**) and mohangiol (**2**).

All ^1^*J*_CH_ correlations were fully assigned based on the HSQC spectroscopic data (Table 1). Analysis of the COSY and HMBC spectra revealed two partial structures (Figure 2). Initially, COSY correlations between H-2 (*δ*_H_ 6.39) to H-3 (*δ*_H_ 7.71) confirmed a direct connection between the sp^2^ carbons C-2 (*δ*_C_ 125.1) and C-3 (*δ*_C_ 143.2). Further analysis of COSY correlations of olefinic protons between H-3-H-4 (*δ*_H_ 6.43)-H-5 (*δ*_H_ 6.45) resulted in the construction of a diene moiety spanning from C-2 to C-5 (*δ*_C_ 129.4). The large ^1^H-^1^H coupling constants (*J* = 15.5 Hz) observed for the H-2 and H-3 olefinic protons and ROESY signals between H-2 and H-4, and H-3 and H-5 allowed the assignment of the double bond configurations as 2*E* and 4*E*. The COSY correlation between H-5 and H-6 (*δ*_H_ 2.69) revealed that the diene moiety was directly connected to C-6 aliphatic carbon (*δ*_C_ 41.2). Additional COSY spectroscopic signals from H-6 to H_2_-12 (*δ*_H_ 3.53, 3.19) extended the chain from C-6 to C-12 (*δ*_C_ 43.2). Three methyl groups (C-14, C-15, C-16; *δ*_C_ 18.4, 16.2, and 14.7) were located at C-6, C-8 (*δ*_C_ 34.6), and C-10 (*δ*_C_ 37.3), respectively, based on homonuclear correlations between H_3_-14 (*δ*_H_ 1.17)/H-6, H_3_-15 (*δ*_H_ 1.18)/H-8, and H_3_-16 (*δ*_H_ 1.19)/H-10 (*δ*_H_ 2.07). The ^2^*J*_CH_ couplings from H_2_-2 to C-1 (*δ*_C_ 171.7) and from H_2_-12 to C-13 (*δ*_C_ 199.7) established the first partial structure as a 16-carbon chain with three branched methyl groups.

Additionally, ^1^H NMR signals at H-2′/H-6′ (*δ*_H_ 8.26/*δ*_H_ 8.26) and H-3′/H-5′ (*δ*_H_ 8.08/*δ*_H_ 8.08) of mohangic acid H (**1**) displayed characteristic features of para-substituted aromatic ring protons. Strong ^1^H-^1^H homonuclear NMR correlations between H-2′/H-6′ and H-3′/H-5′, along with HMBC correlations from these protons, confirmed the presence of an aromatic ring. ^1^H-^13^C heteronuclear NMR signals from H-2′/H-6′ to C-13 further established connectivity between the first partial carbon chain and the aromatic ring. Additional HMBC correlations from 4′-NH group (*δ*_H_ 11.13) to C-3′/C-5′ (*δ*_H_ 119.2) and C-7′ (*δ*_C_ 169.8) identified an *N*-acetyl functional group at C-4′. Finally, HMBC signals from the singlet methyl proton (*δ*_H_ 2.18) to C-7′ and C-4′, supported by ROESY correlations between H_3_-8′ and 4′-NH group occupied the last open position deduced from the molecular formula, thus establishing the planar structure of **1** (Figure 2A).

Mohangiol (**2**) was purified as a yellow gum determined to possess a molecular formula of C_25_H_40_NO_4_ on the basis of HR-ESIMS data combined with ^1^H and ^13^C NMR data (Table 1). The 1D and 2D NMR spectroscopic data of **2** showed features analogous to those of **1,** with key differences including the absence of a *N*-acetyl group (*δ*_H_ 2.18; *δ*_C_ 24.4 and 169.8) in **1** and the presence of a primary amine, 4′-NH_2_ (*δ*_H_ 6.61). Furthermore, the terminal carboxylic acid group present in **1** was replaced by a secondary alcohol in **2**. NMR data confirmed the extension of the aliphatic chain through methylene carbons (C-3; *δ*_C_ 40.3/C-4; *δ*_C_ 30.0). Consequently, mohangiol (**2**) was determined to be a derivative bearing a carbon backbone similar to that of the previously reported mohangic acid G, wherein the terminal carboxylic acid moiety was replaced by a secondary alcohol (Figure 2B).

In a previous study, mohangic acids A–E were completely degraded under the MTPA chemical derivatization conditions. Thus, the absolute configurations of mohangic acid H (**1**) and mohangiol (**2**) were determined based on the comparison of their ^13^C NMR chemical shifts to those of previously reported mohangic acids A–E and their presumed common origin from the same biosynthetic pathway, establishing the absolute configurations as 6*R*, 7*S*, 8*R*, 10*R*, 11*R* in **1** and 9*R*, 10*S*, 11*R*, 13*R*, 14*R* in **2**. However, unlike the terminal carboxylic acid group reported in other mohangic acids, mohangiol (**2**) possesses a secondary alcohol group at the C-2 position, introducing a new stereogenic center. To address this, a quantum-mechanics-based chemical shift analysis method, DP4+ calculations [14], was employed. The two possible diastereomers, **2a** (2*R*) and **2b** (2*S*), were proposed (Figure 3A). Then, the ^1^H and ^13^C chemical shifts of a total of 9 conformers were calculated and averaged by their Boltzmann populations. The comparison of the experimental chemical shifts and the Boltzmann-averaged chemical shifts suggested the 2*S* configuration with 100% probability (Figure 3B, Appendix A).

### 2.2. Proposed Biosynthetic Pathway of Mohangic Acids

To elucidate the biosynthetic pathway of mohangic acids and their new derivatives, whole-genome sequencing of *Streptomyces* sp. AWH31-250 was performed, generating a 7.35 Mb draft assembly (Appendix A). Analysis using antiSMASH 8.0 [15] identified 22 biosynthetic gene clusters (BGCs) encoding polyketides, non-ribosomal peptides, and other secondary metabolites (Appendix A). Among these BGCs, a 271 kb type I PKS-NRPS hybrid gene cluster (region 2.1) was found, displaying a similarity confidence score of 0.93 to the candicidin biosynthetic gene cluster predicted as putative mohangic acids and their new derivatives cluster (Figure 4A, Appendix A) [16,17].

The six PKS genes, *moh*A–F, encode 21 modules, comprising one loading and 21 elongation modules. The *p*-aminobenzoic acid (PABA) starter unit was loaded by *mohP1* and *mohP2*, homologs of *PabAB* and *PabC*. *mohP1*, which contains glutamine amidotransferase and chorismate-binding motifs, catalyzed the conversion of chorismate and glutamine to 4-amino-4-deoxychorismate (ADC). Then, *mohP2* converted ADC into PABA and pyruvate. Activated PABA was loaded onto the carboxylic acid–CoA ligase (CoL) and acyl carrier protein (ACP) domains of *mohA*. Mature polyketide chain elongation and tailoring reactions were synthesized by elongation modules encoded in *mohA–F* (homologous to *FscA–F*), and the thioesterase (TE) domain finally releases the completed backbone, establishing the core scaffold of the mohangic acids and candicidin (Figure 4B) [18,19,20,21].

Polyketide chain release to generate mohangic acids A–G and their new derivatives (**1** and **2**) may be catalyzed by *mohC* PKS genes. Mohangic acid G, with the shortest polyketide chain, was released from module 6 and *mohF4* (acetyltransferase) induced *N*-terminal acetylation, producing mohangic acid H (**1**). Mohangic acid F was released from module 7 with inactivation of dehydratase (DH). As a result of release from module 8, mohangic acid A is presumed to be the main precursor, generating mohangic acid B, C, and E via the stand-alone domain *mohF2* (methyltransferase), *mohF4* (acetyltransferase), and *mohF3* (glycosyltransferase). Involvement of *mohF1*, short-chain dehydrogenase/reductase (SDR), reduced ketone group in mohangic acid C to hydroxyl group, producing mohangic acid D. Notably, decarboxylation by *mohF5* and subsequent reduction of a conjugated double bond by the SDR-family reductase *mohF1* of mohangic acid A generated the first terminal alcohol-bearing derivative, mohangiol (**2**) (Figure 4B) [22,23,24,25,26]. The *mohC* includes modules 5–11, releasing at module 8 to produce mohangic acids and extending to module 11 to yield the candicidin biosynthetic pathway.

### 2.3. Bioinformatic Analysis for Stereochemical Assignment

Further bioinformatic sequence analysis was performed to support the absolute configuration of the C-6, C-7, C-8, C-10, C-11 in **1** and C-9, C-10, C-11, C-13, C-14 in **2**. In general, polyketide synthase (PKS) contains *α*-, *β*-, *γ*-, *δ*-module structures using malonyl-CoA and methylmalonyl-CoA as building blocks. Especially, ketoreductase (KR) in *β*-module has been reported to determine the absolute configuration of *β*-hydroxy group, displaying six *β*-A1, *β*-A2, *β*-B1, *β*-B2, *β*-C1, *β*-C2 KR subtypes (Appendix A) [27,28,29]. Within the BGC of mohangic acid H and mohangiol, the first *β*-ketoreductase (KR) in module 2, *mohB*_KR1, contains a representative motif 2 (LDD) for B-type configurations in the absence of motif 9 (W) for A-type configurations. Additional motif 3 (R) and motif 5 (T) were conserved in the sequence of *mohB*_KR1, supporting *mohB*_KR1 as B-type configurations. Further bioinformatic analysis identified that the representative motif 10 (P) for B2-type configuration was replaced as “A” in *mohB*_KR1 and motif 3 (R/Q) was conserved in this sequence, suggesting *mohB*_KR1 is classified as a B1-type configuration. Unfortunately, B1-type KR module does not have an epimerization function at the *α*-carbon; the absolute configuration of C-10 and C-11 based on genomic analysis does not match with our previous report [11].

Also, *mohB*_KR2 and *mohB*_KR3, which determine the absolute configuration of C-6, 7, and 8, were classified as *γ*-module (KS-AT-DH-KR-ACP) and *δ*-module (KS-AT-DH-ER-KR-ACP), respectively. Due to the complicated allosteric interaction between KR and dehydratase (DH), it is hard to predict stereochemical assignment based on KR fingerprint motifs [29]. Thus, we analyzed the genomic homology of all KR domains between *mohB*–*C* and *FscB*–*C* to compare the biosynthetic pathway of candicidin and its absolute configuration (Appendix A). Surprisingly, *β*-module *FscB*_KR2 conserved representative motifs for the B1-type configuration, and this result also does not match with the absolute configuration of candicidin (Figure 5). Additional KR in candicidin biosynthetic gene cluster also contains identical fingerprint motifs with KR in mohangic acid biosynthetic gene cluster, supporting the limitation of bioinformatic analysis in the determination of the absolute configuration of compounds of this structural class. Nevertheless, the Takahashi group reported total synthesis of mohangic acid C in 2022, revealing chemical shifts at C-8, C-10, and C-11 that partially differed from those of natural mohangic acid C [30]. Based on these findings, further studies involving bioinformatic analysis combined with total synthesis are needed to confirm the absolute configurations of the stereogenic centers at C-8, C-10, and C-11 in mohangic acids.

### 2.4. Biological Evaluation

The biological activities of mohangic acid H (**1**) and mohangiol (**2**) were evaluated in multiple ways. **1** and **2** were evaluated for inhibitory activities against pathogenic bacteria (*Staphylococcus aureus* ATCC 25923, *Enterococcus faecalis* ATCC 19433, *Enterococcus faecium* ATCC 19434, *Salmonella enterica* ATCC 14028, *Escherichia coli* ATCC 25922, *Klebsiella pneumoniae* ATCC 10031) and fungi (*Candida albicans* ATCC 10231, *Aspergillus fumigatus* HIC 6094, *Trichophyton rubrum* NBRC 9185, *Trichophyton mentagrophytes* IFM 40996) in broth microdilution MIC assays. No significant antimicrobial effects were detected (MIC > 128 µM; Appendix A).

Given their proposed role as biosynthetic precursors of candicidin, both **1** and **2** were next assessed for inhibition of *Candida albicans* isocitrate lyase (ICL), a key enzyme in the glyoxylate cycle essential for fungal virulence under nutrient limitation [31,32]. ICL cleaves d-isocitrate into succinate and glyoxylate and plays a crucial role in enabling *Candida albicans* to survive in nutrient-limited host environments and maintain infection [33]. Given its absence in mammals, ICL represents a promising target for selective antifungal therapy, and inhibition of this enzyme is considered an effective strategy that impairs pathogen survival and virulence [34]. As hypothesized, mohangic acid H (**1**) and mohangiol (**2**) exhibited moderate ICL inhibition (IC_50_ = 21.37 and 21.12 µg/mL, respectively), approximately 1.4-fold less potent than the positive control 3-nitropropionate (IC_50_ = 15.02 µg/mL; Figure 6, Table 2).

## 3. Materials and Methods

### 3.1. General Experimental Procedures

Optical rotations were measured using a JASCO P-200 polarimeter (JASCO, Easton, PA, USA) with a sodium light source and a 1 cm cell. UV spectra were acquired on a PerkinElmer Lambda 35 UV/Vis spectrophotometer (Perkin Elmer, Waltham, MA, USA). IR spectra were recorded using a Thermo Nicolet iS10 detector (Thermo Fisher Scientific, Waltham, MA, USA). ^1^H, ^13^C, and 2D NMR spectra were obtained using Bruker Avance 600 MHz NMR spectrometers (Bruker BioSpin, Billerica, MA, USA) at the National Center for Inter-University Research Facilities (NCIRF) at Seoul National University. Electrospray ionization mass spectrometry (ESI–MS) data were acquired on an Agilent 1260 Infinity HPLC system coupled to a G6125C single quadrupole mass spectrometer (Agilent Technologies, Santa Clara, CA, USA). Chromatographic separation was achieved on an Agilent Eclipse Plus C_18_ column (150 × 4.6 mm, 5 µm) with a suitable gradient elution. Ultra-high performance liquid chromatography (UHPLC) was conducted on a Revident LC/Q-TOF with 1290 Infinity III platform (Agilent Technologies, Santa Clara, CA, USA) using an Eclipse Plus C_18_ column (Agilent Technologies, Santa Clara, CA, USA, 100 × 2.1 mm, 1.8 µm).

### 3.2. Bacterial Isolation

A bacterial strain AWH31-250 was isolated from a mud sample obtained from the sanctuary of migratory birds at Nakdong River estuary, Republic of Korea. Geographically, it was located at 35°04′51″ N, 128°54′44″ E (Appendix A). The sample was air-dried at room temperature for 9 h. Two isolation methods were employed: direct stamping of dry sediment and spreading of sediment suspension (1 g sediment in 4 mL sterilized artificial sea water). The sample was inoculated onto various selective media including HV agar (750 mL artificial sea water, 250 mL distilled water, 18 g agar, 1 g humic acid, 0.5 g Na_2_HPO_4_, 1.7 g KCl, 50 mg MgSO_4_·7H_2_O, 10 mg FeSO_4_·H_2_O, 20 mg CaCO_3_, 1 mL vitamin B solution), AS medium (750 mL artificial sea water, 250 mL distilled water, 18 g agar, 5 mg/L streptomycin), chitin-based agar (750 mL artificial sea water, 250 mL distilled water, 18 g agar, 4 g chitin powder, 3.5 g K_2_HPO_4_, 0.75 g MgSO_4_·7H_2_O, 3.5 g KH_2_PO_4_, 10 mg FeSO_4_·H_2_O, 10 mg MnCl_2_·4H_2_O, 10 mg ZnSO_4_·7H_2_O), SC medium (750 mL artificial sea water, 250 mL distilled water, 18 g agar, 10 g starch, 1 g casein, 0.5 g K_2_HPO_4_), and 10% YPG medium with cycloheximide (750 mL artificial sea water, 250 mL distilled water, 18 g agar, 0.4 g yeast extract, 0.4 g peptone, 0.8 g glucose (Junsei Chemical Co., Ltd., Tokyo, Japan), 50 mg/L cycloheximide). Artificial sea water was prepared by Instant Ocean^®^ sea salt (Spectrum Brands, Blacksburg, VA, USA). Media components were obtained from Sigma-Aldrich (St. Louis, MO, USA) and BD Difco™ (Franklin Lakes, NJ, USA) unless otherwise noted. The strain AWH31-250 was isolated from HV agar plate.

### 3.3. Identification and Classification of the Strain

The 16s rRNA sequencing analysis conducted at SolGent Co., Ltd. (Daejon, Republic of Korea) revealed that strain AWH31-250 (GenBank accession no. PP863879.1) shares the highest sequence similarity (99.93%) with *Streptomyces lividans*, identifying it as a member of *Streptomyces* sp. The 16s rRNA gene sequence (Appendix A) of strain AWH31-250 was manually aligned with the top 20 sequences from the NCBI BLAST 2.17.0+ (National Center for Biotechnology Information, Bethesda, MD, USA) results using MEGA 11.0 software (MEGA software, Philadelphia, PA, USA), and phylogenetic trees were constructed using the neighbor-joining, maximum-likelihood, and maximum-parsimony methods. The resulting tree topologies were validated by bootstrap analysis based on 500 resampled datasets using MEGA 11.0. The strain was positioned within a subclade that included the type strains of *Streptomyces lividans*, *Streptomyces daghestanicus*, and other *Streptomyces* species (Appendix A). This relationship was consistently supported by all tree-making algorithms and demonstrated a high bootstrap value, corresponding to a single nucleotide difference at position 1192 out of 1399 nt.

### 3.4. Cultivation and Extraction

The strain AWH31-250 was grown in 50 mL of YEME liquid medium (4 g yeast extract (BD Difco™, Franklin Lakes, NJ, USA), 10 g malt extract (BD Difco™, Franklin Lakes, NJ, USA), and 4 g glucose (Junsei Chemical Co., Ltd., Tokyo, Japan) in 750 mL artificial sea water and 250 mL distilled water) in a 125 mL Erlenmeyer flask. After incubation for 3 days, on a rotary shaker (Daihan Scientific Co., Wonju, Republic of Korea) at 180 rpm at 30 °C, 10 mL of the culture was transferred directly to 125 mL of YEME liquid medium in a 250 mL Erlenmeyer flask. After fermentation for 3 days, under the same conditions, 25 mL of the culture was transferred directly to 1.5 L of YEME liquid medium in 3 L Erlenmeyer flask with a baffled Erlenmeyer flask (63 each × 1.5 L, total volume 94.5 L). After the bacteria were cultured for 8 days, the 94.5 L culture of the AWH31-250 strain was extracted with 150 L of EtOAc (Samjun Scientific Co., Ltd.™, Seoul, Republic of Korea). The EtOAc layer was collected, and anhydrous sodium sulfate (Samjun Scientific Co., Ltd.™, Seoul, Republic of Korea) was added to remove residual water. The extract in EtOAc was evaporated in vacuo (EYELA N-1300, Tokyo Rikakikai Co., Ltd., Tokyo, Japan) to yield 10 g of dried material in total (Appendix A).

### 3.5. Isolation of Mohangic Acid H (***1***) and Mohangiol (***2***)

The organic extract from microbial strain AWH31-250 was first subjected to open column chromatography on a YMC*GEL ODS-A (75 µm, 12 nm, YMC Co., Ltd., Kyoto, Japan) using a gradient solvent system of 20%, 40%, 60%, 80%, and 100% MeOH/H_2_O. Fractions corresponding to 60% and 80% MeOH/H_2_O eluents were collected. These fractions were then filtered with a syringe filter and directly injected into a semi-preparative reversed-phase HPLC column (Kromasil C_18_ (2): 250 × 10 mm, 5 μm, Nouryon, Bohus, Sweden). A gradient solvent system of 20% MeOH/H_2_O to 80% MeOH/H_2_O over 60 min was applied, with detection at 280 nm (flow rate: 2 mL/min). Two major peaks were observed at retention times of 42 min and 46 min. Each compound was further purified using gradient solvent conditions (30% CH_3_CN/H_2_O to 80% CH_3_CN over 40 min, UV detection at 280 nm, flow rate: 2 mL/min) on a reversed-phase C_18_ HPLC column (Kromasil C_18_ (2): 250 × 10 mm, 5 μm). This process yielded pure compounds, identified as mohangic acid H (**1**) and mohangiol (**2**), with retention times of 19.2 min (3.5 mg) and 23.8 min (4.2 mg), respectively (Appendix A). All solvents were sourced from Honeywell Specialty Chemicals (Morristown, NJ, USA).

Mohangic acid H (**1**): Yellow gum; ${\left[\alpha \right]}_{D}^{25}$ = −12.6 (*c* 0.5, MeOH); UV (MeOH) *λ*_max_ (log *ɛ*) 275 (4.32) nm, IR (neat) *ν*_max_ 3395, 2961, 1724, 1566, 1525 cm^−1^; ^1^H and ^13^C NMR data, Table 1; HR-ESIMS *m*/*z* 432.2372 [M + H]^+^ (Calcd. for C_24_H_34_NO_6_, 432.2386; Appendix A).

Mohangiol (**2**): Yellow gum; ${\left[\alpha \right]}_{D}^{25}$ = –18.4 (*c* 0.5, MeOH); UV (MeOH) *λ*_max_ (log *ɛ*) 235 (4.57) nm, 315 (4.26) nm; IR (neat) *ν*_max_ 3410, 2964, 1602, 1546, 1395 cm^−1^; ^1^H and ^13^C NMR data, Table 1; HR-ESIMS *m*/*z* 418.2962 [M + H]^+^ (Calcd. for C_25_H_40_NO_4_, 418.2957; Appendix A).

### 3.6. Conformational Search and DP4+ Analysis

To identify the stereostructure of mohangiol (**2**), computational DFT (density functional theory) calculation was performed. The initial structural energy minimizations of **2** were established by using Avogadro 1.2.0 with UFF. A conformational search of mohangiol (**2**) was conducted using the Material Science program (Version 5.2, Schrödinger LLC, New York, NY, USA) integrated within Maestro (Version 13.8, Schrödinger LLC). During this search, conformers of diastereomers for two models within a 10 kJ/mol energy window were identified based on the MMFF (Merck Molecular Force Field) method. These conformers underwent geometry optimization using the B3LYP/6-31+G(d,p) level of density functional theory (DFT) within the polarizable continuum model (PCM) to account for solvent effects. To predict the NMR chemical shifts, the shielding tensor values for protons and carbons ($\sigma^{0}$) were calculated for the optimized conformers at the B3LYP/DFT level. The calculated chemical shift for each nucleus ($\delta_{calc}^{x}$) was determined using the shielding tensor equation. These values were averaged based on the Boltzmann population, derived from the associated Gibbs free energy of the conformers. The resulting averaged NMR shifts were subsequently used for DP4+ analysis. This analysis was performed using an Excel sheet provided by the original developers of the DP4+ methodology, allowing for a probabilistic assignment of stereochemical configurations based on the agreement between experimental and calculated NMR data.$$\delta_{calc}^{x}=\frac{\sigma^{0}-\sigma^{x}}{1-\sigma^{0}/10^{6}}$$

### 3.7. Sequencing and Gene Annotation

The whole genome sequencing data of mohangic acids producing *Streptomyces* sp. AWH31-250 was acquired by the Department of Molecular Bioscience, Kangwon National University. Bacterial genomic DNA extraction was performed by adding a lysozyme treatment for 2 h to the CTAB extraction protocol [35]. Genome sequencing was performed using Illumina NovaSeq and Nanopore platform of R10.4.1 version. Illumina library was made with Illumina Nextera DNA Sample Preparation Kit (Illumina Inc., San Diego, CA, USA) according to the manufacturer’s instructions, followed by 100 bp paired-end sequencing. Nanopore library was prepared using Ligation Sequencing Kit V14 (SQK-LSK114). Genome assembly of the Illumina and Nanopore sequencing data was accomplished with Unicycler v0.5.0 [36]. Genome annotation was accomplished by prokka [37], and prediction of biosynthetic gene clusters was performed using antiSMASH 8.0 [15].

### 3.8. Bioinformatic Analysis

The ketoreductase (KR) sequences of *mohB* and *mohC* were identified and extracted using antiSMASH 8.0 [15], while homologous domains from *FscB* and *FscC* were retrieved from the MIBiG database [38]; representative *β*-module KR subtypes (*β*-A1, *β*-A2, *β*-B1, *β*-B2, *β*-C1, *β*-C2) were compiled based on previous studies and the additional MIBiG database. The multiple sequence alignments were conducted in MEGA 12 (MEGA software, Philadelphia, PA, USA) [39] using MUSCLE algorithm [40], and conserved fingerprint motifs (motifs 1–10) were visualized as sequence logos with WebLogo 3.7.9 (University of California, Berkeley, CA, USA) [41]. All alignments and logos were manually inspected to ensure accurate motif calling and KR subtype classification.

### 3.9. Antibacterial Activity Assay

Three species of Gram-positive bacteria (*Staphylococcus aureus* ATCC 25923, *Enterococcus faecalis* ATCC 19433, and *Enterococcus faecium* ATCC 19434) and three species of Gram-negative bacteria (*Klebsiella pneumoniae* ATCC 10031, *Salmonella enterica* ATCC 14028, and *Escherichia coli* ATCC 25922) as human pathogens were selected as test strains and prepared for the antibacterial activity assays. The test bacterial strains were cultured overnight in Mueller–Hinton broth (MHB) at 37 °C for human pathogens and in tryptic soy broth (TSB) at 28 °C for *B. thuringiensis*. The grown strains were harvested by using a centrifuge and washed twice with sterilized water. Ampicillin and tetracycline were used as positive controls, and dimethyl sulfoxide (DMSO) was used as a negative control. Each compound, including **1** and **2**, was dissolved in DMSO and diluted with MHB or TSB to prepare serial two-fold dilutions (from 128 µg/mL to 0.015 µg/mL). For the antibacterial activity assay, 10 µL of suspensions containing 1 × 10^6^ colony forming units (CFU)/mL of the test bacteria and 190 µL of MHB or TSB, which included the positive control compounds or test compounds, were added into each well of a 96-well plate (final concentration of test bacteria was 5 × 10^4^ CFU/mL). Then, the plate was incubated at 37 °C or 28 °C for 24 h. The MIC values of each compound were defined as the lowest concentration at which the compound inhibited the growth of the test bacteria.

### 3.10. Antifungal Activity Assay

Potato dextrose agar (PDA) was used to cultivate *Candida albicans* ATCC 10231. After incubation for 48 h at 28 °C, the yeast cells were harvested by centrifugation and washed twice with sterile distilled water. *Aspergillus fumigatus* HIC 6094, *Trichophyton rubrum* NBRC 9185, and *Trichophyton mentagrophytes* IFM 40,996 were also plated on PDA and incubated for 2 weeks at 28 °C. Spores were harvested and washed twice with sterile distilled water. Stock solutions of **1** and **2** were prepared in DMSO. Each stock solution was diluted with RPMI 1640 broth (Difco, Livonia, MI, USA) to give serial twofold dilutions in the range of 0.06–128 µg/mL. The final DMSO concentration was maintained at 1% by adding DMSO to the broth. Aliquots (10 µL) of the RPMI 1640 broth containing approximately 10^4^ cells/mL were mixed with the test compound solutions in each well of a 96-well plate. The plates were incubated for 24 h (for *C. albicans*), 48 h (for *A. fumigatus*), and 96 h (for *T. rubrum* and *T. mentagrophytes*) at 37 °C. A culture with DMSO (1%) was used as a solvent control, and a culture supplemented with amphotericin B was used as a positive control.

### 3.11. Isocitrate Lyase (ICL) Activity Assay

The reaction mixture (1 mL) comprised 20 mM sodium phosphate buffer (pH 7.0), 1.27 mM *threo*-dl-(+)-isocitrate, 3.75 mM MgCl_2_, 4.1 mM phenylhydrazine, and 2.5 µg/mL recombinant ICL. The reaction was initiated immediately by the addition of substrate, in the presence or absence of a specified concentration of inhibitor prepared in DMSO (final concentration, 0.5%). Formation of glyoxylate phenylhydrazone was monitored spectrophotometrically at 324 nm following incubation at 37 °C for 30 min. Protein concentrations were quantified using the Bradford method with the Bio-Rad protein assay kit (Bio-Rad Laboratories, Inc., Hercules, CA, USA), employing bovine serum albumin as a standard. An inhibitor-free control was incorporated, and the percentage inhibition of ICL enzyme activity by each compound was calculated relative to this control. 3-nitropropionate served as a positive control, inhibiting ICL with an IC_50_ value of 15.02 µg/mL.

## 4. Conclusions

Two new *p*-aminoacetophenone derivatives, mohangic acid H (**1**) and mohangiol (**2**), were isolated from *Streptomyces* sp. AWH31-250. Comprehensive 1D/2D NMR, high-resolution MS, UV spectroscopy, and quantum mechanics-based DP4+ calculations established their planar and absolute structures. Although the candicidin biosynthetic gene cluster was identified in 2003, its precursor metabolites involved in this biosynthetic gene cluster have not been found for 22 years. Especially, the first plausible biosynthesis of mohangic acids was proposed in this study, providing genetic evidence that mohangic acid H (**1**) and mohangiol (**2**) serve as candicidin precursors. Furthermore, the inhibitory activity of *p*-aminoacetophenone derivatives against *Candida albicans* isocitrate lyase has not been reported before, connecting mohangic acid H (**1**) and mohangiol (**2**) to candicidin, a potent antibiotic against *Candida albicans*. Our results not only highlight the importance of exploring microorganisms from extreme marine environments to discover novel bioactive compounds, but also suggest new molecular templates possessing an alternative antifungal mechanism targeting the glyoxylate cycle.

## Figures and Tables

**Figure 2 marinedrugs-23-00307-f002:**
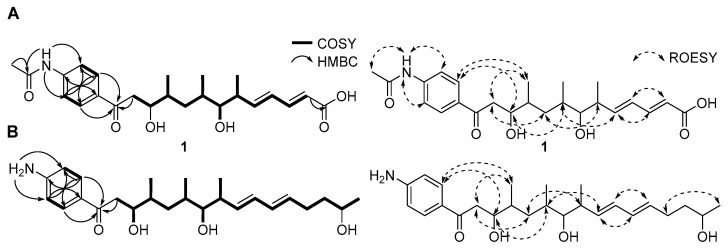
Key COSY, HMBC, and ROESY correlations for constructing the planar structures of (**A**) mohangic acid H (**1**) and (**B**) mohangiol (**2**).

**Figure 3 marinedrugs-23-00307-f003:**
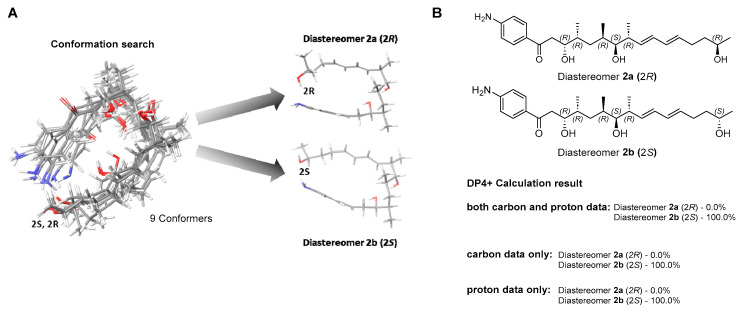
Conformational ensemble and DP4+ probability analysis of mohangiol (**2**). (**A**) Overlay of low-energy conformers generated by MMFF-based search and optimized at the B3LYP/6-31+G(d,p) level in a PCM; two representative conformer types are highlighted. (**B**) DP4+ probability assignment based on combined ^13^C/^1^H, ^13^C-only, and ^1^H-only NMR data.

**Figure 4 marinedrugs-23-00307-f004:**
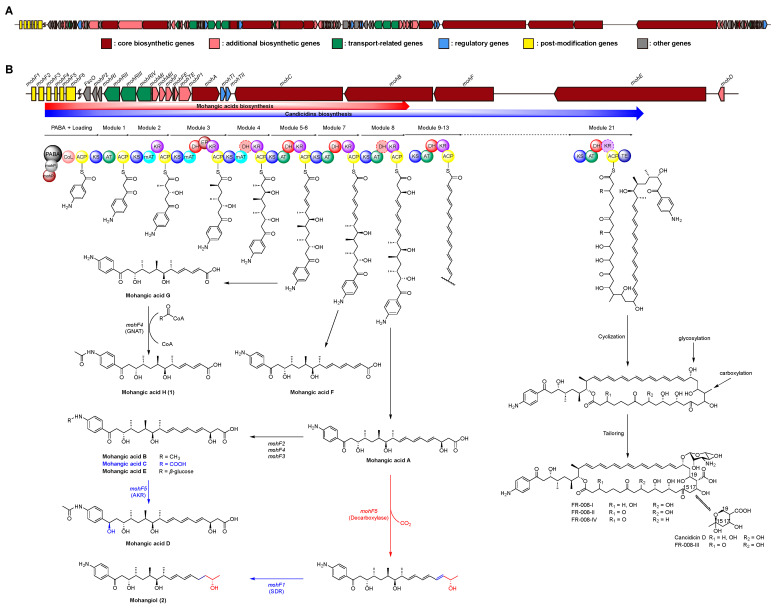
(**A**) Biosynthetic gene cluster on region 2.1 of *Streptomyces* sp. AWH31-250. (**B**) Biosynthetic gene cluster for mohangic acids and their new derivatives (**1** and **2**), as well as candicidin and its derivatives, with a schematic representation of the predicted by circles. The dash circle indicates a domain that was predicted not to be active. Biosynthetic pathway of mohangic acids and their derivatives was part of the biosynthetic pathway of candicidin.

**Figure 5 marinedrugs-23-00307-f005:**
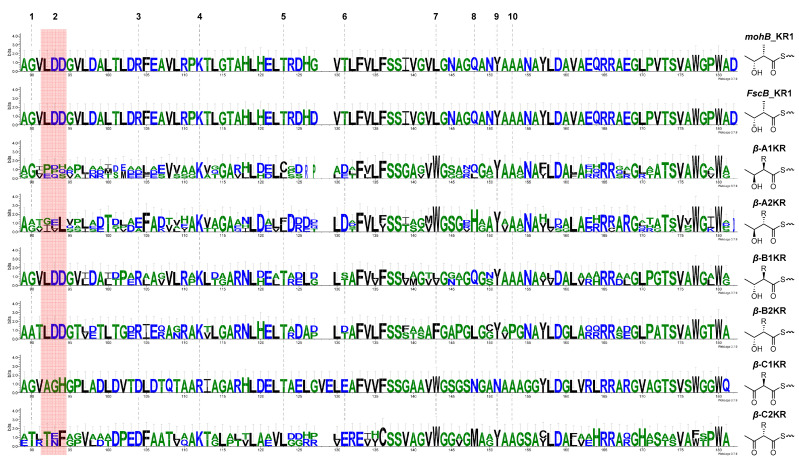
Sequence logo-based comparison of core fingerprint motifs in *β*-module ketoreductase (KR) domains, aligning *mohB*_KR1 and *FscB*_KR1 with representative KR subtypes (*β*-A1, *β*-A2, *β*-B1, *β*-B2, *β*-C1, *β*-C2).

**Figure 6 marinedrugs-23-00307-f006:**
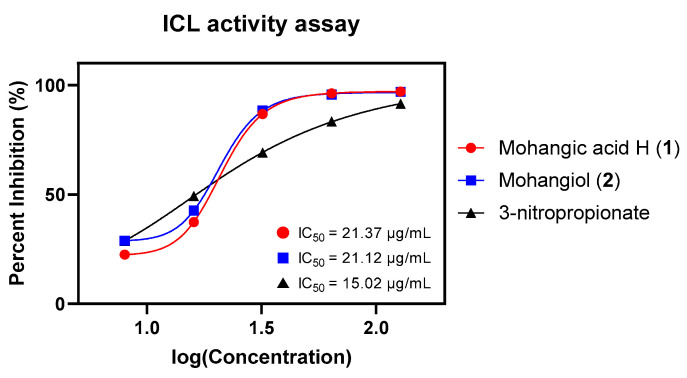
Dose-response curves and IC_50_ values of mohangic acid H (**1**), mohangiol (**2**), and 3-nitropropionate on ICL inhibitory activity test. 3-nitropropionate was used as a standard inhibitor.

**Table 1 marinedrugs-23-00307-t001:** ^1^H and ^13^C NMR data of mohangic acid H (**1**) and mohangiol (**2**) in pyridine-*d*_5_.

	Mohangic Acid H (1) ^a^				Mohangiol (2) ^a^	
No.	*δ*_C,_ Type	*δ*_H,_ Mult (*J* in Hz)		No.	*δ*_C,_ Type	*δ*_H,_ Mult (*J* in Hz)
1	171.7, C			1	24.7, CH_3_	1.35, d (6.0)
2	125.1, CH	6.39, d (15.5)		2	66.8, CH	4.05, br. s
3	143.2, CH	7.71, dd (15.5, 11.0)		3a	40.3, CH_2_	1.64, m
				3b		1.79, m
4	145.4, CH	6.43, m		4a	30.0, CH_2_	2.34, m
				4b		2.41, m
5	129.4, CH	6.45, m		5	132.9, CH	5.75, m
6	41.2, CH	2.69, m		6	131.8, CH	6.22, dd (15.0, 10.5)
7	79.1, CH	3.5, dd (9.0, 5.0)		7	131.3, CH	6.28, dd (15.0, 10.5)
8	34.6, CH	1.95, m		8	136.2, CH	6.03, m
9a	36.7, CH_2_	1.65, m		9	41.2, CH	2.65, m
9b		1.85, m				
10	37.3, CH	2.07, m		10	79.8, CH	3.49, m
11	72.8, CH	4.62, m		11	34.5, CH	2.00, m
12a	43.2, CH_2_	3.53, dd (15.0, 9.0)		12a	37.3, CH_2_	1.70, m
12b		3.19, dd (15.5, 3.0)		12b		1.88, m
13	199.7, C			13	37.6, CH	2.09, m
14	18.4, CH_3_	1.17, d (6.5)		14	73.3, CH	4.63, br. s
15	14.7, CH_3_	1.19, d (6.5)		15a	42.9, CH_2_	3.19, d (15.5)
				15b		3.46 (m)
16	16.2, CH_3_	1.18, d (6.5)		16	198.7, C	
				17	18.9, CH_3_	1.20, d (6.5)
				18	14.7, CH_3_	1.19, d (6.5)
				19	16.4, CH_3_	1.21, d (6.5)
1′	133.6, C			1′	127.7, C	
2′	130.3, CH	8.26, d (8.5)		2′	131.9, CH	8.18, d (8.5)
3′	119.2, CH	8.08, d (8.5)		3′	114.0, CH	6.95, d (8.5)
4′	145.1, CH			4′	154.8, CH	
5′	119.2, CH	8.08, d (8.5)		5′	114.0, CH	6.95, d (8.5)
6′	130.3, CH	8.26, d (8.5)		6′	131.9, CH	8.18, d (8.5)
4′-NH		11.13, s				
				4′-NH_2_		6.61, s
7′	169.8, C					
8′	24.4, CH_3_	2.18, s				

^a 1^H and ^13^C NMR data were recorded at 600 and 150 MHz, respectively.

**Table 2 marinedrugs-23-00307-t002:** Results of isocitrate lyase (ICL) activity assay of **1** and **2**.

Sample	128 µg/mL	64 µg/mL	32 µg/mL	16 µg/mL	8 µg/mL	IC_50_ (µg/mL)
Mohangic acid H (**1**)	97.01%	96.22%	86.76%	37.35%	22.49%	21.37
Mohangiol (**2**)	96.89%	95.68%	88.29%	42.54%	28.90%	21.12
3-nitropropionate *	91.40%	83.41%	69.03%	49.28%	28.76%	15.02

* 3-nitropropionate was used as a standard inhibitor of ICL.

## Data Availability

All data is contained within this article and Appendix A.

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
