# Peer review of "Mohangic Acid H and Mohangiol: New p-Aminoacetophenone Derivatives from a Mudflat-Derived Streptomyces sp."

_marinedrugs, 2025, doi:10.3390/md23080307_

Round 1

Reviewer 1 Report

Comments and Suggestions for Authors

This article describes the isolation, biological activity and biosyntesis of two new members of the mohangic acid family. although the structural novelty of the compounds is not high, the fact that their biosynthetic gene cluster has been identified and their inhibition of isocitrate lyase, comparable to that of the control 3-nitropropionate merit their publication in Marine Drugs. 

However, some changes are considered necessary to be implemented prior to publication of the article. These are the following:

1) The stereochemistry of the compounds has been established based on biosynthetic considerations and similarity of the NMR spectra with those of other members of the family. In the opinion of this reviewer, the absolute configuration of the molecules must be secured via ECD analysis. Alternatively, having the BGC of the compounds identified, perhaps an analysis of the ketoreductase modules generating stereochemistry should be performed according to the procedure described in the following article: https://doi.org/10.1016/j.chembiol.2007.07.009

2) In table 1, including the NMR data, revise the following points:

a) Revise the multiplicity and coupling constants assigned to H-7 in compound 1.

b) Reconsider the multiplicity and coupling constants assigned to H-6 an H-7 in compoound 2. A coupling constant of 12.5 Hz soule not be compatible with the E configuration assigned to both double bonds. Perhaps a reanalysis of the signals around 6.25 ppm would allow the assignment of two double doublets. If not, please describe the signals as multiplest due to overlapping.

c) Revise the coupling constants of H-9 in compound 2. A coupling constant of 15.5 hZ is not compatible with the structure of the compound

d) Finally consider acquiring 1D TOCSY esperiments irradiating H-6, H-8, and H10 in 1, and H-9, H-11, and H-13 in 2, which would allow the determination of the coupling constants of H-14, H-15 and H-16 in 1, and of H-17, H-18, and H-19 in 2

3) The units of the values in table 2 must be indicated.

4) Please note that Figure S1C does not contain the LC/MS chemical profile of comounds 1 and 2, but therir UV and mass spectra.

5) Other changes to be implemented are included in the editec copy of the document provided

Author Response

I appreciate your valuable comments. Please see the attachment. 

Reviewer 2 Report

Comments and Suggestions for Authors

The manuscript “Mohangic acid H and Mohangiol: New p-Aminoacetophenone
Derivatives from a Mudflat-Derived Streptomyces sp.” describes the discovery of two new natural compounds - mohangic acid H and mohangiol - isolated from a Streptomyces sp. found in a tidal mudflat in Korea. The authors have determined the chemical structures of these compounds through various spectroscopic methods and computational analysis. They've also proposed a biosynthetic pathway and tested the compounds' biological activities against Candida albicans isocitrate lyase.

The manuscript can be accepted after revisions.

Part 2.1. Line 67: Why only 2 carbonyl carbons, the authors should add carrbonyl ketone signal.

Rewrite sentence line 71-73: “Since ten olefinic carbon signals (corresponding to five double
bonds) along with two carbonyl carbons account for seven degrees of unsaturaion, the structure of mohangic acid H (1) should incorporate additional structure elements to account for the remaining two degrees”.

Line 127: The authors should discuss the absolute configurations of  C-6,7,8, 10, 11 in compound 1 and of C-9, 10,11, 13, 14 in compound 2 first, before mention to C-2 position of compound 2.

Provide the 1H NMR of compounds 1 and 2 with integration, chemical shifts.

In the HSQC and HMBC of compound 1, the 13C NMR spectra was not shown

Author Response

(The authors gave the same response as above.)

Reviewer 3 Report

Comments and Suggestions for Authors

The authors present the isolation, elucidation, and proposed biosynthesis of two natural products related to the Mohangic acid class of compounds. While the structure elucidation seems sound, there are issues with assignment of configuration. For much of the structures, the authors assumed the configuration was the same as published by Oh et al. However, the authors did not cite an article that describes the synthesis of the proposed structure of Mohangic Acid C (Org. Lett. 2022, 24, 18, 3416-3420). Notably, the NMR data for the synthetic compound do not match those reported by Oh et al. It is not surprising that there are issues with the original configurational assignments due to H14-H15 having a large coupling constant and application of J-based configurational analysis of C14-C15. Oh et al. relied on roesy in a non-constrained system to define that relationship since J-based analysis for systems with anti-protons will not work. This aligns well with the differences highlighted in the paper describing the synthesis.

Since the authors have the BGC, it would be worthwhile checking to see if the configuration can be assigned based on rules developed by Adrian Keatinge-Clay's group (reviewed Nat. Prod. Reports, 2016, 33, 141-149). Some of the nomenclature (A-type and B-type) has been included in AntiSMASH for the keto-reductase types that dictate configuration of hydroxyl groups. The authors have a good opportunity to help clarify the configuration of these compounds.

Some clarification regarding configuration needs to be included before publication.

Author Response

(The authors gave the same response as above.)

Round 2

Reviewer 1 Report

Comments and Suggestions for Authors

All my concerns regarding the previous version of the article have been satisfactorily answered by the authors. It is important to note that, although the bioinformatic analysis has not allowed to confirm the absolute configuration of the molecules, it is also of value to report that this kind of analysis might not work in all cases.

I only suggest changing the following text in lines 212-216: "Additional KR in candicidin biosynthetic  gene cluster also contains identical fingerprint motifs with KR in mohangic acid biosyn thetic gene cluster, supporting the limitation of bioinformatic analysis for KR fingerprint motif. Therefore, further studies for the determination of the absolute configuration of mohangic acids may be required."

to be replaced by "Additional KR in candicidin biosynthetic  gene cluster also contains identical fingerprint motifs with KR in mohangic acid biosynthetic gene cluster, supporting the limitation of bioinformatic analysis in the determination of the absolute configuration of compounds of this structural class.

Author Response

I appreciate your valuable comments. We corrected the text in lines 212-216 as you suggested. Please find the revised file and check green-marked sentences. 

Reviewer 3 Report

Comments and Suggestions for Authors

The authors have addressed my concerns. Notably, the analysis of the BGC to assist with configuration was excellent.

Author Response

I appreciate your valuable comments. Due to your review, our manuscript has improved.